# Influence of Cigarette Smoke Inhalation on an Autogenous Onlay Bone Graft Area in Rats with Estrogen Deficiency: A Histomorphometric and Immunohistochemistry Study

**DOI:** 10.3390/ijms20081854

**Published:** 2019-04-15

**Authors:** Camilla Magnoni Moretto Nunes, Daniella Vicensotto Bernardo, Camila Lopes Ferreira, Mônica Fernandes Gomes, Andrea Carvalho De Marco, Mauro Pedrine Santamaria, Maria Aparecida Neves Jardini

**Affiliations:** 1Department of Diagnosis and Surgery, São Paulo State University (UNESP), School of Sciences and Technology, São José dos Campos 12245-000, Brazil; camilla.moretto@unesp.br (C.M.M.N.); daniella.bernardo@unesp.br (D.V.B.); camila.ferreira@unesp.br (C.L.F.); andrea.marco@unesp.br (A.C.D.M.); mauro.santamaria@unesp.br (M.P.S.); 2Department of Bioscience and Bucal Diagnosis, São Paulo State University (UNESP), School of Sciences and Technology, São José dos Campos 12245-000, Brazil; monica.gomes@unesp.br

**Keywords:** bone regeneration, smoking, estrogen deficiency

## Abstract

Purpose: The present study aimed to evaluate the influence of cigarette smoke inhalation on an autogenous onlay bone graft area, either covered with a collagen membrane or not, in healthy and estrogen-deficient rats through histomorphometry and immunohistochemistry. Materials and Methods: Sixty female rats (Wistar), weighing 250–300 g, were randomly divided and allocated into groups (either exposed to cigarette smoke inhalation or not, ovariectomized and SHAM). After 15 days, the test group underwent cigarette smoke inhalation. Sixty days after exposition, autogenous bone grafting was only performed on all right hemimandibles, and the left ones underwent autogenous onlay bone grafting with the collagen membrane (BioGide^®^). The graft was harvested from the parietal bone and attached to the animals’ jaws (right and left). They were euthanized at 21, 45, and 60 days after grafting. Histological measurements and immunohistochemical analyses were performed, and results were submitted to a statistical analysis. Results: The addition of a collagen membrane to the bone graft proved more efficient in preserving graft area if compared to the graft area without a collagen membrane and the one associated with cigarette smoke inhalation at 21 (*p* = 0.0381) and 60 days (*p* = 0.0192), respectively. Cigarette smoke inhalation combined with ovariectomy promoted a significant reduction of the autogenous graft area at 21 and 60 days. At 45 days, no statistically significant results were observed. In the immunohistochemical analysis, the ovariectomized and smoking subgroups, combined or not with collagen membrane, received moderate and intense immunolabeling at 21 days for Receptor Activator of Nuclear Factor Kappa-B Ligand (RANKL) (*p* = 0.0017 and *p* = 0.0381, respectively). For Osteoprotegerin (OPG), intense immunolabeling was observed in most subgroups under analysis at 60 days. Conclusion: Smoking inhalation promoted resorption on the autogenous onlay bone graft, mainly when associated with ovariectomy. Furthermore, when associated with the collagen membrane, a lower resorption rate was observed if compared to the absence of the membrane.

## 1. Introduction

A common therapeutic challenge in the daily practice of periodontics and implantodontics is bone loss, on account of being a condition that can be the result of periodontal and peri-implant diseases, trauma, anatomical or congenital factors, exodontia, and the use of total or partial dentures, which may promote a continuous reabsorption of alveolar ridges, thus making adequate posterior rehabilitation difficult [1].

Seeking adequate aesthetic and functional rehabilitation, osseointegrated implants have emerged as a safe therapeutic approach with high success rates. However, there must be a minimum amount of bone area, both in width and height [1,2].

Due to such needs, techniques for bone defect repair in the alveolar ridge have been increasingly proposed in the literature [2,3,4,5,6,7].

Guided bone regeneration (GBR) is a technique that has been used in rehabilitation cases with dental implants in which there is an insufficient amount of bone area in the graft bed [8,9,10]. However, in order to be successful, the technique requires: A proper selection of the type of graft and its mechanical stabilization, prevention of bacterial infection, conservation of the area under the membrane, separation of osteogenic cells from non-osteogenic ones [1,11], membrane stability, peripheral sealing between the membrane and bone, and adequate blood supply [12]. In addition, the patient’s systemic factors must also be taken into account, since they may interfere with the osseointegration process and compromise the treatment [13].

An example of such factors is smoking, associated with a consequent high morbidity index [14,15], and alveolar bone loss [16]. Its pathophysiological effects affect arteriolar vasoconstriction, cellular hypoxia, bone demineralization, and delayed revascularization [17,18,19,20]. Thus, it is noted a certain difficulty in repairing bone grafts [21,22], a lower success rate of titanium implants, greater bone loss around previously osseointegrated implants [23,24], reduction in collagen production, and impaired function of polymorphonuclear leukocytes and macrophages [19].

As smoking, osteoporosis is another important systemic condition that affects bone graft repair and is a progressive disease which is characterized by bone fragility and mass reduction, thus altering architecture and compromising its resistance, and leading to a greater propensity of bone fractures at minimal trauma [25,26].

Osteoporosis combined with smoking may promote an increase of bone fractures risk [27,28], by systemically affecting bone remodeling through Receptor Activator of Nuclear Factor Kappa-B Ligand/Osteoprotegerin (RANKL/OPG) signaling during events that modulate osteoclast cell differentiation, the formation of interleukin IL-1 and interleukin IL-6, and bone resorption of cytokines at high levels [29]. The cumulative effect of smoking generates direct toxicity on bone mineral density [30].

Therefore, scientific studies conducted in animal models can provide important information about the mechanisms of action in biological processes by providing answers to the proposed interventions. Studies in the literature on the influence of treatments, such as Guided Bone Regeneration (GBR), and risk factors on bone tissue used in animal tests, especially in rats. Thus, this animal model can be used to verify and understand the complex interaction of important factors and the treatment options on bone tissue.

Given the increased demand for prosthetic rehabilitation with osseointegrated implants and reconstructive procedures, and due to the need for studies evaluating the interaction of such conditions, the present study aimed to evaluate the influence of cigarette smoke inhalation on an autogenous onlay bone graft repair covered, or not, for the collagen membrane in the jaw of healthy and estrogen-deficient rats through histomorphometry and immunohistochemistry.

## 2. Results

### 2.1. Descriptive Histology and Statistical Analysis

The histological and statistical comparisons between ovariectomized subgroups revealed for the non-smoking, ovariectomized and collagen membrane subgroup (COM), less resorption of the lateral borders of the graft compared to the ovariectomized and smoking subgroup (TO) at 21 days (COM × TO, *p* = 0.0381) and also at 60 days (COM × TO, *p* = 0.0192). The bone graft lodged in the receiver bed by the dense vascularized connective tissue and covered by the collagen membrane was observed in the COM subgroup, in both periods. Instead, the TO subgroup observed a poor integration of the bone graft into the receiver bed, with the presence of a loose connective tissue interposed between the receiver bed interface where there was no integration. The association of cigarette smoke with ovariectomy had significantly reduced the autogenous onlay bone graft area at 21 days (COM × TO) (Figure 1, Figure 2 and Figure 3). The results were statistically significant when compared to ovariectomized animals, although they were not exposed to cigarette smoke inhalation with a collagenous membrane covering the graft, mainly at 60 days (COM × TO) (Figure 2).

The membrane associated with the autogenous bone graft proved more efficient in conserving the graft area in comparison with the bone graft with no membrane (CSM × CS, *p* = 0.0010 at 21 days). The CSM subgroup, even as COM, was lodged to the receiver bed surrounded by the dense vascularized connective tissue and the collagen membrane, whereas CS subgroup presented the graft integration however, there was a slight resorption of the graft edges at 21 days. The same situation was noted for the graft associated with cigarette smoke inhalation (CSM × CS TS, *p* < 0.0001) at 21 days, in which partial graft integration was observed in subgroup TS with a slight resorption of the graft area and the presence of loose connective tissues at the point where there was no integration to the receiver bed.

The association of cigarette smoke inhalation with the collagen membrane led to significant results in graft area conservation when compared to smoking associated with the graft (TSM × TS, *p* < 0.0001) and control group (TSM × CS, *p* = 0.0087) at 21 days. In TSM, the main difference between the histological characteristics observed in the TS and CS subgroups was the bone neoformation characterized by the presence of immature and trabecular bone tissues and mature bone tissues on the receiver bed (Figure 1 and Figure 3).

The membrane–graft association resulted in statistically significant results at 60 days in comparison with control subgroups (CSM × CS CS, *p* = 0.0154). Although in the CSM, the graft presented was most integrated to the receiver bed, but with bone neoformation and a slight resorption at the graft periphery in both subgroups. The CSM subgroup also presented statistically significant results compared to the smoke associated with the graft subgroup (CSM × CS TS, *p* = 0.003), in which the bone graft showed to be poorly integrated into the bed–graft interface, and it was noted the presence of loose connective tissues interposed between regions where there was no integration, in addition to resorption of lateral borders of the graft. Finally, CSM presented statistically significant when compared to the smoking–graft membrane (CSM × CS TSM, *p* = 0.0190) in which the bone graft was slightly reabsorbed at the edges, and partly integrated into the receiver bed with the presence of areas of loose connective tissues interposed between the bed–graft interface where there was no integration (Figure 2 and Figure 3).

For all the groups in a 45-day period, no statistically significant differences were observed (Figure 3).

### 2.2. Immunohistochemical Assessment

#### 2.2.1. Receptor Activator of Nuclear Factor Kappa-B Ligand (RANKL)

In the CO subgroup, weak immunolabeling was most prevalent in all periods and compared to the TOM and TO (intense, *p* < 0.0001, for both subgroups), was statistically significant at 21 days (Figure 4).

In the COM subgroup, moderate immunolabeling was observed at 45 days, but weak at 21 days. It was statistically significant compared to TOM (intense, *p* = 0.0017) and TO (intense, *p* = 0.0381) at 21 days, and to TOM (weak, *p* = 0.0088) and TO (weak, *p* = 0.0088) at 45 days (Figure 4 and Figure 5).

In the TS subgroup, intense immunolabeling was observed at 21 days and compared to CSM and CS (weak, *p* < 0.0001, for both subgroups) and TSM (moderate, *p* < 0.0001) was statistically significant (Figure 4 and Figure 5).

#### 2.2.2. OPG

In the COM subgroup, intense immunolabeling was observed and compared to CO (weak, *p* = 0.0049), TOM (weak, *p* = 0.00198), and TO (weak, *p* = 0.00198), and was statistically significant at 21 days. At 45 days, the COM subgroup (weak), when compared to CO (weak, *p* < 0.0001), TO (weak, *p* = 0.0002), and TOM (moderate, *p* < 0.0001) showed significant statistical differences. In this period, TOM (moderate) compared to TO (weak, *p* = 0.0002) was statistically significant (Figure 4 and Figure 6).

The TSM subgroup (moderate immunolabeling) compared to CS and TS (weak, *p* = 0.0383, for both subgroups) showed statistical differences at 21 days and at 60 days compared to CS (moderate, *p* = 0.0007) and CSM (weak, *p* = 0.0007). The CS subgroup (moderate immunolabeling) compared to CSM and TSM (weak, *p* = 0.0383, for both subgroups) showed significant statistical differences (Figure 4 and Figure 6).

The TSM subgroup (moderate immunolabeling) compared to TS (intense immunolabeling, *p* = 0.0006) showed significant statistical differences at 60 days (Figure 4 and Figure 6).

## 3. Discussion

In this study, the presence of loose connective tissues and reduction of bone neoformation at the bed–graft interface was observed in all subgroups submitted to inhalation of cigarette smoke, which corroborates with the findings of a study by Bonfante et al. (2008) [31]. In the histomorphometric results, the inhalation of cigarette smoke impaired the maintenance of the bone graft area, but when associated with the resorbable collagen membrane, the TSM subgroup presented statistically significant results of maintenance of the area when compared to the TS subgroup at 21 days.

The literature data have indicated that the use of a collagen membrane prevents graft resorption, thus practically keeping its original size [1,3,4,32,33,34,35] as shown in the present study. Sculean et al. (2008) [36] also observed that the use of a collagen membrane associated with bone graft promoted superior tissue repair, rather than grafts without it. Our results agree since subgroup CSM proved more efficient at conserving graft area if compared to subgroups CS and TS at 21 and 60 days.

Histologically, this approach has been proved effective in graft area conservation by the presence of vascularized dense connective tissues and bone neoformation at the bed–graft interface in the CSM subgroup and COM, TSM, and TOM subgroups, thus corroborating literature data [1,12,37,38,39,40,41].

However, according to a study by Gielkens et al., (2008) [42] the results showed that the use of membranes to prevent bone remodeling with reabsorption and to increase the incorporation of autogenous bone grafts is debatable. In addition, in the study, no difference between the types of membranes used in the block grafts and the control group (without membrane coating) was observed. These results confront the data from our study.

Luize et al. (2008) [5] performed a histomorphometric study on onlay autogenous bone grafts without the use of the resorbable collagenous membrane in ovariectomized rats at periods of 7, 14, and 28 days and concluded that estrogen deficiency due to ovariectomy delayed the graft repair process to the receiver bed. Although there was a delay in this process, there were indications that this event could be completed over longer periods. Similarly, our histomorphometric results showed that for the CO subgroup in all the evaluated periods, there was the integration of the graft into the receiver bed. Nascimento et al. (2009) [32] evaluated onlay autogenous bone grafts in ovariectomized rats with or without resorbable collagen membrane at periods of 21, 45, and 60 days and showed that ovariectomy did not negatively affect graft integration, as observed in our results in the periods of 21 and 60 days, in which ovariectomy had not negatively interfered with integration, thus not being a crucial factor in bone repair [33,43,44]. In addition, the area of grafts covered by the collagen membrane presented better maintenance in comparison with subgroups without it, as predicted by Donos et al. (2002) [2], in which the use of a membrane covering the bone graft assisted in greater migration of osteogenic cells, bone neoformation and consequently, in the tissue mineralization process.

A study by Carvalho et al. (2006) [45] elucidated the mechanisms of interaction between cigarette and estrogen deficiency on bone tissue, that resulted in the inhibition of cell proliferation and differentiation in osteoblasts, fibronectin concentration, and platelet-derived growth factor that are a chemoattractant for osteoprogenitor cells and osteoblasts. The data from the TO subgroup resulted in a slight integration of the graft with the bed and the presence of reabsorption of lateral borders, which corroborates the findings of a work conducted by Yuhara et al. (1999) [16], in which osteoblast differentiation in the cell culture was affected by the presence of nicotine, thus exerting a critical effect on bone metabolism. Analyzing the results of the TOM subgroup at 60 days we noted that the association between collagen membrane, smoking, and estrogen deficiency had conserved the graft area in a statistically significant manner compared to TOM at 21 days. This was in agreement with results obtained by Saldanha et al. (2004) [46], in which nicotine affected the healing process of critical defects treated by the ROG, although the process was not impeded by it, and with those obtained by Machado et al. (2010) [47] in which it emphasized an alteration in bone trabeculation and neoformed connective tissue in the presence of nicotine, especially in the early stages of healing, thus delaying the repair process.

Osteoblasts secrete OPG, a soluble receptor that blocks the RANK/RANKL interaction through its binding to RANKL, thereby preventing osteoclast differentiation and activation [48]. Thus, the balance between RANKL and OPG determines osteoclasts’ formation and activity [49].

Smoking may cause changes in osteogenesis, including changes in the balance of RANK/RANKL/OPG [50]. Smokers have a greater potential for bone resorption by increasing IL-1, IL-6, and Tumor Necrosis Factor-Alfa (TNF-α), which stimulate RANKL receptor expression [51].

Estrogen deficiency systematically affects bone remodeling through RANKL/OPG signaling during events that modulate osteoclast cell differentiation and the development of lymphocytes [43,44]. After menopause, bone loss is related to the preference of RANKL activity over OPG. RANKL is the main stimulant for the differentiation, development, maturation, activation, and survival of osteoclasts [34].

The results for the TO and TS subgroups, at 21 days, showed a greater immunolabeling for RANKL which was an activator of osteoclastic-genesis. The results suggest a homeostatic imbalance in the RANK/RANKL/OPG system that induce greater bone resorption.

By analyzing the immunolabeling for OPG, it was observed at 60 days that COM, TOM, CS, CSM, and TS subgroups presented intense immunolabeling. A reduction in estrogen level decreased OPG activity and increased RANKL activity, thus leading to increased bone resorption and loss. However, literature presents conflicting results regarding the postmenopausal period when bone loss favors RANKL activity in comparison with OPG [33,34].

Although histomorphometric and immunolabeling differences were observed between the analyzed subgroups, comparisons of these results with human and other animal models should be made with caution. As observed by Carvalho et al. (2006) [45], differences in the exposure regime to cigarette smoke and the hormonal profile of each study model should also be considered. In addition, there is no species of ideal animal model in studies on bone tissue [52] and new experimental methodologies with modifications in the membrane properties aiming at bone regeneration [53] are necessary for a better understanding of the results. Despite these limitations, our results suggest hypotheses to better guide the understanding of biological events on the bone graft in animals subjected to the analyzed conditions. Smoking promoted damage in the autogenous bone graft repair mainly when combined with ovariectomy, however, in the limitations of this study, these conditions did not prevent the repair process from occurring. Furthermore, when associated with the collagen membrane, a lower resorption rate was observed if compared to the absence of the membrane.

## 4. Materials and Methods

This study was approved by the Local Research Ethics Committee at the São Paulo State University (UNESP), Institute of Science and Technology, São José dos Campos, protocol 10/2012-PA/CEP (14 December 2012; Ethics Committee on Animal Research) and the Animal Research: Reporting of In Vivo Experiments guidelines (ARRIVE) guidelines for reporting in vivo animal experiments were followed. Sixty female adult Wistar (*Rattus norvegicus*, *Albinus*) rats (weighing between 250 and 300 g) were randomized and allocated to experimental groups, a control group (C—animals with no cigarette smoke inhalation exposure), and a test group (T—animals exposed to cigarette smoke inhalation). The former group was ovariectomized (O) and the latter underwent an ovariectomy simulation (S), resulting in eight subgroups. For each animal in all groups, the left hemi-mandible received the bone graft with the collagenous membrane (M) (BioGide^®^, Geistlich, REF: 30802.6) and the right hemi-mandibles only received the bone graft.

Throughout the experimental period, the animals remained in plastic cages, identified according to the respective subgroup, and were fed on a regular diet with water ad libitum by a qualified staff at the animal house from the Institute of Science and Technology, São José dos Campos.

A sample size equal to 5 (animals) and standard deviation of 0.3 units (estimated value by pilot study), was found using the statistical program PIFACE (Softpedia, Bucharest, Romania) that the power analysis, using the Tukey test (5%) to compare averages, was possible to detect a difference of 0.4 units, for treatment effect with power test up to 80%.

### 4.1. Protocol for Smoking Cigarette Inhalation

The methodology used for such a purpose had been based on previous studies [13,17] which made use of an acrylic box in which 5 animals were passively exposed to the smoke of 10 cigarettes at a time (concentration of 1.3 mg of nicotine, 16.5 mg of tar, and 15.2 mg of carbon monoxide; MINISTER^®^ king size unique; Souza Cruz, Rio de Janeiro, Brazil) 3 times a day for 8 min each. Sixty days after the beginning of cigarette smoke exposure, bone augmentation was performed on the animals’ jaws.

### 4.2. Surgical Procedure

Prior to surgical procedures, the animals were anesthetized with a solution of 13 mg/kg 2-(2,6-xylidine)-5-6-dihydro-4H-1,3-thiazine hydrochloride, a substance with sedative and analgesic properties, as well as a muscle relaxant and 33 mg/kg ketamine base, general anesthetic, administered intramuscularly.

The calvarium was used as a donor site for autogenous bone grafting, and the angle of the jaw was the receiver bed. A single calibrated examiner performed all surgical procedures. A detailed description of part of the procedure on which this work was based can be found in Jardini et al., (2005) and Nascimento et al., (2009) [3,31].

In each calvarium, the bone graft was removed with a trephine drill (4.1 mm in diameter) (Neodent^®^; Curitiba, Brazil) and the center of the removed bone matter was drilled with a helical drill (1.2 mm in diameter, Neodent^®^; Curitiba, Brazil). The receiver bed was also drilled in the same manner so as to allow block stabilization on the receiver bed through a titanium fixation screw with 1.5 mm head diameter, 1.4 mm body diameter, and 2.5 mm length. In the left hemi-mandibles, the graft was covered with the collagen membrane (BioGide^®^, Geistliech, Wolhusen, Switzerland), but only the graft was attached to the receiver bed in the right hemi-mandibles. Wound closure was performed by suturing the muscular layer with a 5-0 polyglactin 910 thread and 4-0 silk thread. After surgery, the animals were fed on a regular diet with water ad libitum and a single dose of an intramuscular antibiotic (1 mg/kg) and a single dose of Ketoprofen (5 mg/kg, Laboratório Teuto Brasileiro S.A.; Anápolis, Brazil) by the subcutaneous route for control of postoperative pain.

### 4.3. Euthanasia and Processing of the Sample

According to the experimental periods of 21, 45, and 60 days, 5 animals from each subgroup were anesthetized and euthanized through cardiac perfusion with 4% formalin. Their hemi-mandibles were removed, cataloged, and fixed in 4% formaldehyde in 0.1 M phosphate buffer solution (pH 7.4) for 48 h, and then demineralized in 10% ethylenediaminetetraacetic solution (EDTA; Dinâmica^®^ Química Contemporânea Ltda.; Diadema, Brazil) for 120 days and were subsequently embedded in paraffin for histological and immunohistological processing. Serial sections were obtained from the central region of the bone graft, but 300 μm from the central portion was occupied by the screw for better structures visualization. For histological and immunohistochemistry procedures sections of 5 μm (*n* = 5) and 3 μm (*n* = 5) thickness, respectively, were obtained.

### 4.4. Histomorphometric Analysis

Routine histological procedures for light microscopy were performed and the samples stained with hematoxylin and eosin (H&E). The histological sections were scanned on a scanner (3DHISTECH), converted into an image by the Pannoramic Viewer 1.15.4 (3DHISTECH), and analyzed quantitatively through the Image J 1.31 software (U.S. National Institutes of Health; Bethesda, MD, USA) by a calibrated and blinded examiner (Pearson 0,97). After calibration, histomorphometric measurement of the graft surface area was calculated (in mm 2) at 25× magnification for all groups, and data collected for statistical analysis.

### 4.5. Immunohistochemistry Analysis

The sections were dewaxed in xylol and alcohol baths. Antigenic recovery was performed in 10 mM sodium citrate solution, pH 6.0, and incubated in a pascal pressure cooker (DAKO, Carpinteria, CA, USA). Endogenous peroxidase blockade occurred in 3% hydrogen peroxide solution for 30 min, followed by washing in PBS associated with the Triton detergent for 5 min. After dilution of the primary antibody in commercial antibody diluent (Spring-cod ADS-125) at 1:400 for OPG (sc–9072, Santa Cruz Biotechnology, Inc, Santa Cruz, CA, USA) and 1:100 for RANK L (sc–7628, Santa Cruz Biotechnology, Inc.) the sections received the solution and were incubated overnight for 18 h at 4 °C. After this time, the excesses of the primary antibody from the slides were removed and washed with PBS. For the amplification of the reaction, the cuts received the horseradish peroxidase HRP-polymer (Nichirei Biosciences Inc.; Tokyo, Japan; cod. 414154F) which was incubated for 30 min in a humid chamber. The slides were again washed in PBS for the reaction developing step. The liquid DAB (3,3–diaminobenzidine) (Spring Bioscience; Fremont, CA, USA; cod DAB-125) was diluted in DAB diluent in the ratio 1:50. This solution was applied over the cuts and allowed to soak for 3 to 5 min and then the slides were counterstained with hematoxylin. The sections were deparaffinized in xylol and rehydrated in alcohol.

For the negative control, one of the cuts was incubated only with the antibody diluent (Spring-cod ADS-125). In other words, the antibody was not included in the procedure. The same other procedures were applied to the protocol. The positive control was the development of a golden-brown precipitate as the final product of the reaction in bone tissue, specifically into osteoblasts.

The immunolabeling intensity was performed according to the semi-quantitative scale for RANKL and OPG, as absence (no immunolabeling), weak (+: 0% to 25%), moderate (++: 25% to 50%), or intense (+++: >50%) [4,54,55,56] for a blinded and calibrated examiner (CMMN). For this analysis, we used 5 sections for the subgroup. In order to facilitate intergroup comparisons, scores of 0, 1, 2, and 3 were attributed to absence, weak, moderate, and intense labeling, respectively, and submitted to statistical analysis [54,56].

### 4.6. Statistical Analysis

Statistical assumptions were evaluated before statistical analysis through the Shapiro–Wilk test and the results indicated that the residuals were normally distributed and, by plotting against predicted values, the uniformity was checked, then none of the ANOVA assumptions were violated. In order to evaluate the relationship between the treatment and sacrifice time, the data obtained in our experiment for both the ovariectomized and SHAM groups were submitted to the two-way statistical analysis of variance model.

The data were analyzed statistically by the analysis of variance (Two-way ANOVA) performed with GraphPad Prism (version 7.00 for Windows, GraphPad Software; La Jolla, CA, USA) and Tukey’s multiple comparison test (α = 5%).

## 5. Conclusions

The results of this animal study showed that smoking inhalation promoted resorption on the autogenous onlay bone graft, mainly when associated with estrogen deficiency. Furthermore, when associated with the collagen membrane, a lower resorption rate was observed if compared to the absence of the membrane.

Thus, the clinical relevance is mainly in the use of the collagenous membrane associated with the bone graft, which has had very positive results in several clinical trials in systemically normal patients and in the present study was also favorable when associated with risk factors, such as smoking and estrogen-deficient conditions commonly seen in adult patients who may need bone reconstruction.

## Figures and Tables

**Figure 1 ijms-20-01854-f001:**
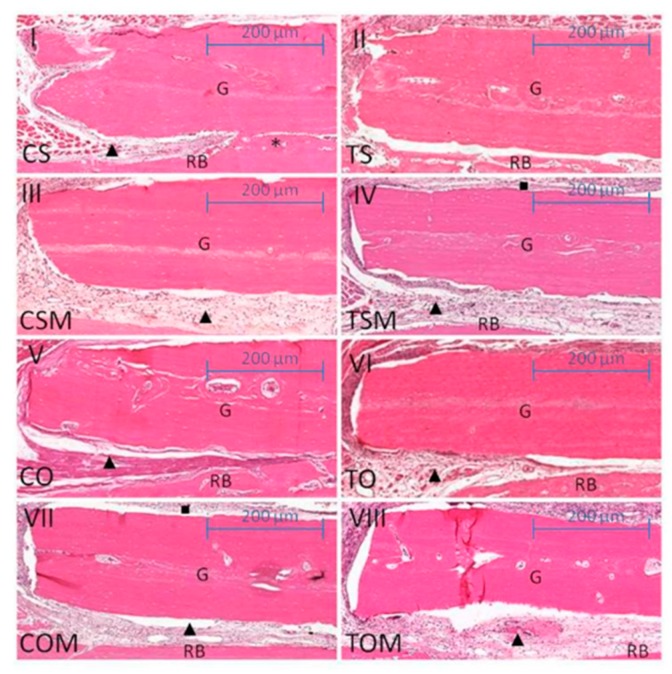
Scanned images for histomorphometric analysis of 21 days. (**I**) subgroup non-smoking and SHAM (CS); (**II**) subgroup smoking and SHAM (TS); (**III**) subgroup non-smoking, SHAM and graft combined with collagen membrane (CSM); (**IV**) subgroup smoking, SHAM and graft combined with collagen membrane (TSM); (**V**) subgroup non–smoking and ovariectomized (CO); (**VI**) subgroup smoking and ovariectomized (TO); (**VII**) subgroup non-smoking, ovariectomized and graft combined with collagen membrane (COM); (**VIII**) subgroup smoking, ovariectomized and graft combined with collagemn membrane (TOM). G = graft; RB = receiver bed; ● integration; ▲ connective tissue; * neoformation; ■ collagen membrane. Scale 200 µm.

**Figure 2 ijms-20-01854-f002:**
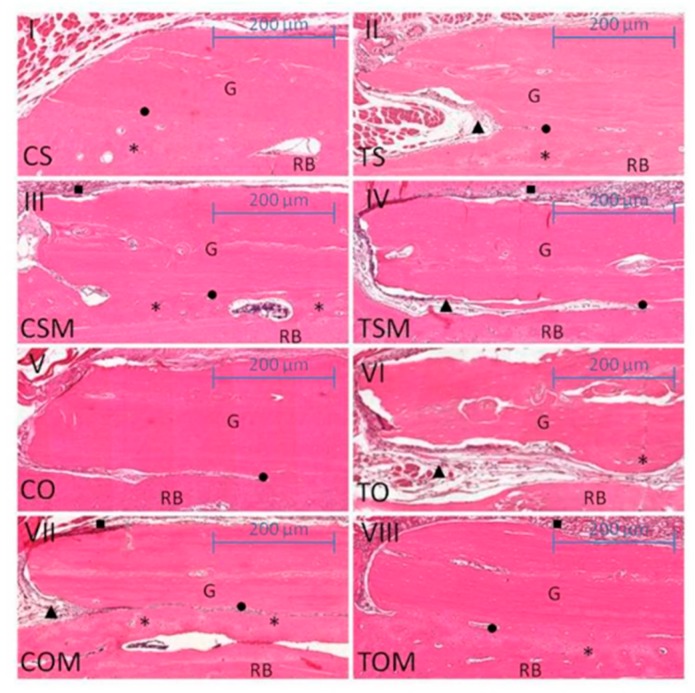
Scanned images for histomorphometric analysis of 60 days. (**I**) subgroup non-smoking and SHAM (CS); (**II**) subgroup smoking and SHAM (TS); (**III**) subgroup non-smoking, SHAM and graft combined with collagen membrane (CSM); (**IV**) subgroup smoking, SHAM and graft combined with collagen membrane (TSM); (**V**) subgroup non–smoking and ovariectomized (CO); (**VI**) subgroup smoking and ovariectomized (TO); (**VII**) subgroup non-smoking, ovariectomized and graft combined with collagen membrane (COM); (**VIII**) subgroup smoking, ovariectomized and graft combined with collagemn membrane (TOM). G = graft; RB = receiver bed; ● integration; ▲ connective tissue; * neoformation; ■ collagen membrane. Scale 200 µm.

**Figure 3 ijms-20-01854-f003:**
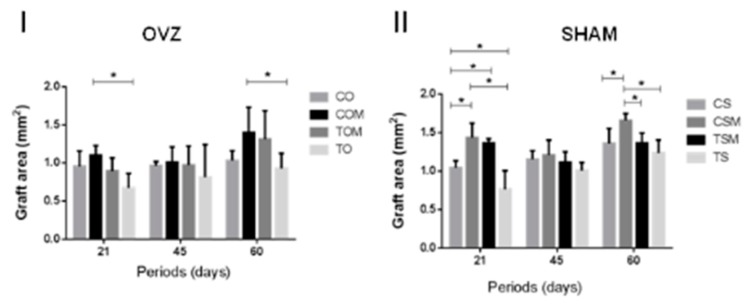
Means and standard deviations for graft area to OVX subgroups (**I**) and SHAM subgroups (**II**). * Statistically significant differences (*p* < 0.05).

**Figure 4 ijms-20-01854-f004:**
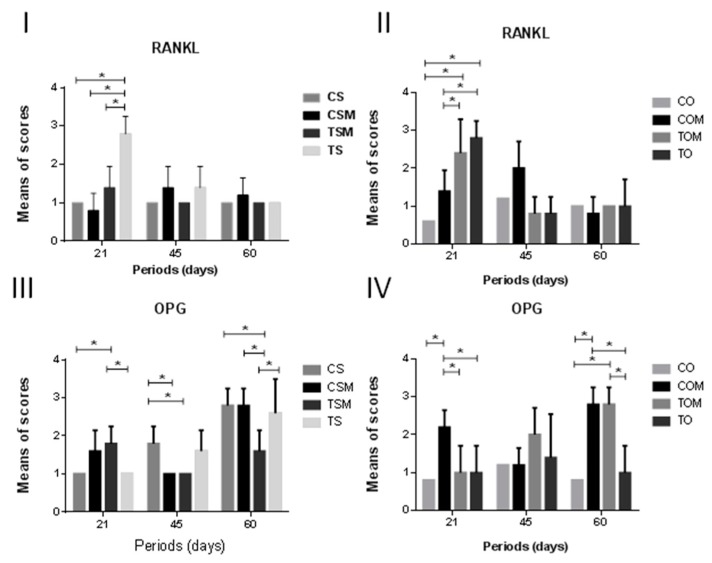
Means and standard deviations for immunolabeling to RANKL (**I**, SHAM; **II**, OVX) and OPG (**III**, SHAM; **IV**, OVX). * Statistically significant differences (*p* < 0.05).

**Figure 5 ijms-20-01854-f005:**
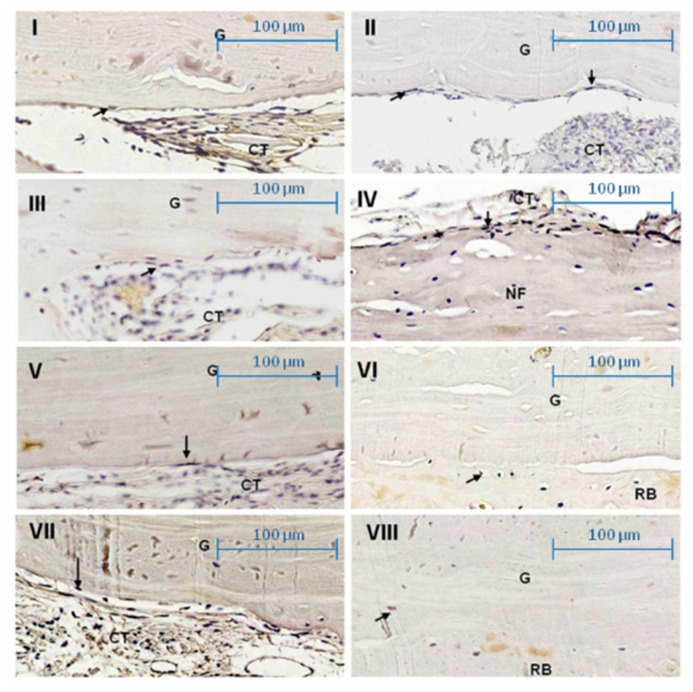
Photomicrographs showing the osteoblasts’ RANKL staining (arrows) on graft surface (G) at ×400 magnification. (**I**) Day 21 Group TOM: Moderate labeling. (**II**) Day 60 Group TOM: Mild labeling of osteoblasts. (**III**) Day 21 Group TO: Intense labeling of osteoblasts on newly formed bone (NF). (**IV**) Day 60 Group TO: Mild labeling of osteoblast. (**V**) Day 21 Group CS: Mild labeling of osteoblast. (**VI**) Day 60 Group CS: Mild labeling of osteoblast. (**VII**) Day 21 Group TS: Intense labeling of osteoblasts. (**VIII**) Day 60 Group TS: Mild labeling of osteoblasts. Receiver bed (RB), connective tissue (CT). Scale 100 µm.

**Figure 6 ijms-20-01854-f006:**
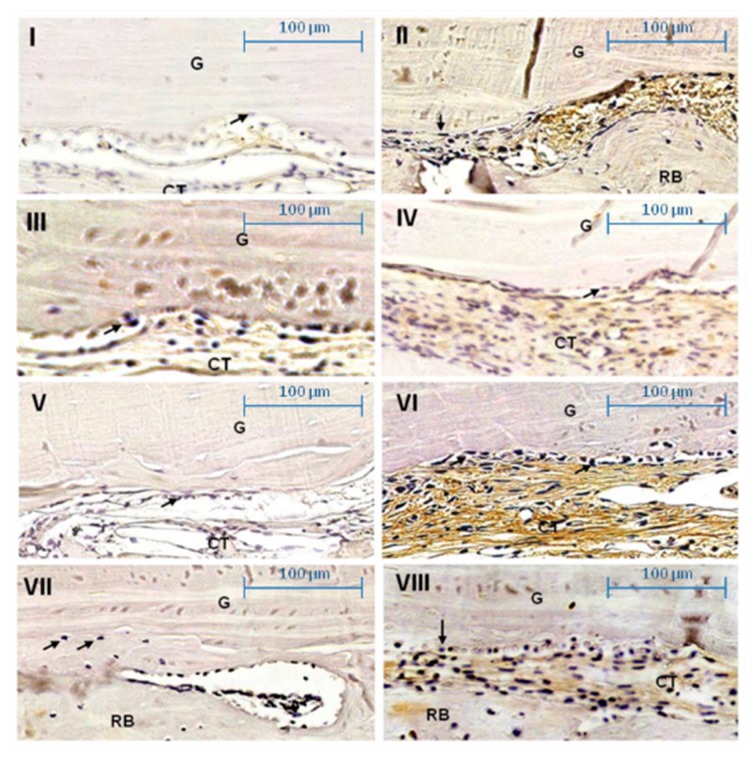
Photomicrographs showing the osteoblast’s OPG staining (arrows) on graft surface (G) at ×400. (**I**) Day 21 Group COM: Moderate labeling. (**II**) Day 60 Group COM: Intense labeling of osteoblasts. (**III**) Day 21 Group TOM: Mild labeling of osteoblasts. (**IV**) Day 60 Group TOM: Intense labeling of osteoblast. (**V**) Day 21 Group CS: Mild labeling of osteoblast. (**VI**) Day 60 Group CS: Intense labeling of osteoblast. (**VII**) Day 21 Group TS: Mild labeling of osteoblasts. (**VIII**) Day 60 Group TS: Intense labeling of osteoblasts. Receiver bed (RB), connective tissue (CT). Scale 100 µm.

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
