# Peer review of "Influence of Cigarette Smoke Inhalation on an Autogenous Onlay Bone Graft Area in Rats with Estrogen Deficiency: A Histomorphometric and Immunohistochemistry Study"

_ijms, 2019, doi:10.3390/ijms20081854_

Round 1

Reviewer 1 Report

I read the authors' responses and I am not satisfied with modifications that they implemented in the revised manuscript for the comments shown below. At this point, I don't have any further comments to add because the authors could not amend the required changes or don't agree with the reviewer's comments. Hence, I will leave the final decision whether to accept or ask for additional changes for the editor or the associate-editor who is in charge of this manuscript.

- The expected positive staining of the IHC was not described in material and methods, nor in the results. Hence, the interpretations of the IHC images are confusing and it is not clear if the arrows are pointing towards positively stained cells or the hematoxylin conter staining of the nuclei. Based on the anatomical structure and location, it seems that the arrows are pointing towards the hematoxylin counterstaining of the nuclei.
Author Answer: This methodology is described briefly in the body text, and the reference was added:
---[I feel the authors didn't read my comment carefully]

- The semi-quantitative scale of the IHC images used by a blinded and calibrated examiner can't be translated into quantitative numbers. This type of analysis could be misleading because IHC is not a linear chemical reaction and can reach a saturation level within a short period of time. The data should be presented similar to the scale used for the evaluation. However, it is important to determine the positive signal based on the staining color by using negative control and sections stained only with hematoxylin. Without these controls, it is hard to understand what the arrows are referring to.
Author Answer: According to Saito CT, Gulinelli JL, Panzarini SR, Garcia VG, Okamoto R, Okamoto T, Sonoda CK, Poi WR. Effect of low level laser therapy on the healing process after tooth replantation: a histomorphometrical and immunohistochemical analysis. Dent Traumatol. 2011 Feb;27(1):30-9. Doi:10.1111/j.1600-9657.2010.00946.x. and Hawthorne AC, Xavier SP, Okamoto R, Salvador SL, Antunes AA, Salata LA. Immunohistochemical, tomographic, and histological study on onaly bone graft remodeling. Part III: allografts. Clin Oral Implants Res. 2013 Oct;24(10):1164-72. doi:10.1111/j.1600-0501.2012.02528.x. is possible the semi-quantitative scale of the IHC images can translated into quantitative number.
----[It is hard to judge the articles that they listed without understanding what they had done. The authors should provide additional information how they did that and what kind of controls (positive and negatives) was used.]

Author Response

I read the authors' responses and I am not satisfied with modifications that they implemented in the revised manuscript for the comments shown below. At this point, I don't have any further comments to add because the authors could not amend the required changes or don't agree with the reviewer's comments. Hence, I will leave the final decision whether to accept or ask for additional changes for the editor or the associate-editor who is in charge of this manuscript.

- The expected positive staining of the IHC was not described in material and methods, nor in the results. Hence, the interpretations of the IHC images are confusing and it is not clear if the arrows are pointing towards positively stained cells or the hematoxylin conter staining of the nuclei. Based on the anatomical structure and location, it seems that the arrows are pointing towards the hematoxylin counterstaining of the nuclei.
Author Answer: This methodology is described briefly in the body text, and the reference was added:
---[I feel the authors didn't read my comment carefully]

Answer: The authors apologize for the wrong impression to the reviewer and will try to answer the questions in a way that is satisfied with the answers:

1-    For the negative control, one of the cuts was incubated only with the antibody diluent (Spring-cod ADS-125) in other words, the antibody was not included in the procedure. The same other procedures were applied, it was washed with PBS and incubated with HRP polymer (Nichirei-cod 414191F - Nichirei Biosciences INC., Tokyo Japan) for 30 minutes at room temperature, in a humid chamber. The slides were again washed in PBS for the developing reaction step. The liquid DAB (SPRING - DAB-125 cod) was diluted in DAB diluent in the 1:50 ratio. This solution was applied over the cuts and allowed to soak for 3 to 5 minutes and then the slides were counterstained with hematoxylin. In summary, as the antibody was not included in the solution, the expected result was a uniform color slide without immunolabeled points.

2-    The positive control was the development of golden-brown precipitate as the final product of the reaction in bone tissue, specifically into osteoblasts.

3-    The figures below represents the immunolabeling for OPG (A – 2,5x black square of the interest area; B – 40x red arrow pointing the immunolabeled cell) and for RANK L (C – 2,5x black square of the interest area; D – 40x red arrow pointing the immunolabeled cells).

- The semi-quantitative scale of the IHC images used by a blinded and calibrated examiner can't be translated into quantitative numbers. This type of analysis could be misleading because IHC is not a linear chemical reaction and can reach a saturation level within a short period of time. The data should be presented similar to the scale used for the evaluation. However, it is important to determine the positive signal based on the staining color by using negative control and sections stained only with hematoxylin. Without these controls, it is hard to understand what the arrows are referring to.

Author Answer: According to Saito CT, Gulinelli JL, Panzarini SR, Garcia VG, Okamoto R, Okamoto T, Sonoda CK, Poi WR. Effect of low level laser therapy on the healing process after tooth replantation: a histomorphometrical and immunohistochemical analysis. Dent Traumatol. 2011 Feb;27(1):30-9. Doi:10.1111/j.1600-9657.2010.00946.x. and Hawthorne AC, Xavier SP, Okamoto R, Salvador SL, Antunes AA, Salata LA. Immunohistochemical, tomographic, and histological study on onaly bone graft remodeling. Part III: allografts. Clin Oral Implants Res. 2013 Oct;24(10):1164-72. doi:10.1111/j.1600-0501.2012.02528.x. is possible the semi-quantitative scale of the IHC images can translated into quantitative number.
----[It is hard to judge the articles that they listed without understanding what they had done. The authors should provide additional information how they did that and what kind of controls (positive and negatives) was used.]

Answer:

1-    The articles included for the authors used a negative control prepared for each specimen using the same method except for the primary antibody, as used in the present study and explained in the answer above.

2-    The authors standardized 5 minutes for the DAB chemical reaction to avoid the saturation. The choice of this reaction time was based on our previous experiments to standardize the protocol used in the study.

The immunolabeling intensity was classified according to the following semi-quantitative scale: (−) absent, (+) weak, (++) moderate, (+++) intense. In order to facilitate intergroup comparisons, scores of 1, 2, 3, and 4 were attributed to absent, weak, moderate, and intense immunolabeling, respectively, as used in our study. Then, the data were statistically analyzed, as another tests used in these studies cited by the authors. The following articles recently published used the same controls and statistical analyzes as used in our study.

Longo M, Gouveia Garcia V, Ervolino E, Ferro Alves ML, Duque C, Wainwright M, Theodoro LH. Multiple aPDT sessions on periodontitis in rats treated with chemotherapy: Histomorphometrical, Immunohistochemical, Imunological and Microbiological Analyses. Photodiagnosis Photodyn Ther. 2018 Nov 20. pii: S1572-1000(18)30139-X. doi: 10.1016/j.pdpdt.2018.11.014.

Theodoro LH, Longo M, Novaes VCN, Miessi DMJ, Ferro-Alves ML, Ervolino E, de Almeida JM, Garcia VG. Low-level laser and antimicrobial photodynamic therapy on experimental periodontitis in rats submitted to chemotherapy by 5-fluorouracil. Support Care Cancer. 2017 Oct;25(10):3261-3271. doi: 10.1007/s00520-017-3738-0.

Reviewer 2 Report

Many thanks for the revision

Author Response

We thank the reviewer for his considerations.

Reviewer 3 Report

Authors made excellent job addressing all the reviewer comments and requests 

Author Response

We thank the reviewer for his considerations.

This manuscript is a resubmission of an earlier submission. The following is a list of the peer review reports and author responses from that submission.

Round 1

Reviewer 1 Report

The authors of this study investigated the influence of cigarette smoke inhalation, estrogen deficiency and collagen membrane on bone integration and regeneration of autogenous bone grafting. The rationale of the study to determining the influence of smoking on the bone grafting and regeneration is very important for clinical application and translational research. Although the study used rats as an in vivo system to address these biological and translational questions, the presented figures and data are not of good quality and it was very hard to judge the interpretation of the results.

Comments to authors:

- More information is needed for the material and method section. Please add the catalog number of the collagen membrane used in the surgery and how it was used in the surgery. 

- A schematic illustration is needed to orient the readers about the order and structure of the performed surgery in order to understand the histological data. 

- The quality and magnification of Figures 1 and 2 are not sufficient to show the information described in the results. It is hard to see the newly formed bone (neo-bone), site of integration, and the resorption of lateral borders of the grafted bone. Higher magnifications and better labeling for each image are needed.

- The expected positive staining of the IHC was not described in material and methods, nor in the results. Hence, the interpretations of the IHC images are confusing and it is not clear if the arrows are pointing towards positively stained cells or the hematoxylin counterstaining of the nuclei. Based on the anatomical structure and location, it seems that the arrows are pointing towards the hematoxylin counterstaining of the nuclei.

- The semi-quantitative scale of the IHC images used by a blinded and calibrated examiner can't be translated into quantitative numbers. This type of analysis could be misleading because IHC is not a linear chemical reaction and can reach a saturation level within a short period of time. The data should be presented similar to the scale used for the evaluation. However, it is important to determine the positive signal based on the staining color by using negative control and sections stained only with hematoxylin. Without these controls, it is hard to understand what the arrows are referring to. 

Author Response

The authors of this study investigated the influence of cigarette smoke inhalation, estrogen deficiency and collagen membrane on bone integration and regeneration of autogenous bone grafting. The rationale of the study to determining the influence of smoking on the bone grafting and regeneration is very important for clinical application and translational research. Although the study used rats as an in vivo system to address these biological and translational questions, the presented figures and data are not of good quality and it was very hard to judge the interpretation of the results.

Answer: The authors thank the reviewer for the considerations and hope that the answers have been satisfactory in clarifying the points raised.

Comments to authors:

- More information is needed for the material and method section. Please add the catalog number of the collagen membrane used in the surgery and how it was used in the surgery. 

Answer: The catalog number of the BioGide collagen membrane is (REF:30802.6 -25x25mm). The membrane-associated graft assembly was fixed by the titanium fixation screw in the receiver bed.

- A schematic illustration is needed to orient the readers about the order and structure of the performed surgery in order to understand the histological data. 

Answer: The schematic illustration was made in order to help readers to understand the histological data.

Figure 1 - View in PDF attached.

- The quality and magnification of Figures 1 and 2 are not sufficient to show the information described in the results. It is hard to see the newly formed bone (neo-bone), site of integration, and the resorption of lateral borders of the grafted bone. Higher magnifications and better labeling for each image are needed.

Answer: The quality of the figures can be  improved, however, the magnification of the figures is at the limit for the visualization of the histological structures in the region of interest. View the schematic illustration above to aid understanding.

- The expected positive staining of the IHC was not described in material and methods, nor in the results. Hence, the interpretations of the IHC images are confusing and it is not clear if the arrows are pointing towards positively stained cells or the hematoxylin conter staining of the nuclei. Based on the anatomical structure and location, it seems that the arrows are pointing towards the hematoxylin counterstaining of the nuclei.

Answer: This methodology is described briefly in the body text, and the reference was added:

 Ferreira CL, Nunes CMM, Bernardo DV, Pedroso JF, Longo M, Santamaria M Jr, Santamaria MP, Jardini MAN. Effect of orthodontic force associated with cigarette smoke inhalation in healthy and diseased periodontium. A histometric and immunohistochemistry analysis in rats. J Periodontal Res. 2018 Oct;53(5):924-931. doi: 10.1111/jre.12584.

- The semi-quantitative scale of the IHC images used by a blinded and calibrated examiner can't be translated into quantitative numbers. This type of analysis could be misleading because IHC is not a linear chemical reaction and can reach a saturation level within a short period of time. The data should be presented similar to the scale used for the evaluation. However, it is important to determine the positive signal based on the staining color by using negative control and sections stained only with hematoxylin. Without these controls, it is hard to understand what the arrows are referring to. 

Answer: According to Saito CT, Gulinelli JL, Panzarini SR, Garcia VG, Okamoto R, Okamoto T, Sonoda CK, Poi WR. Effect of low level laser therapy on the healing process after tooth replantation: a histomorphometrical and immunohistochemical analysis. Dent Traumatol. 2011 Feb;27(1):30-9. Doi:10.1111/j.1600-9657.2010.00946.x. and Hawthorne AC, Xavier SP, Okamoto R, Salvador SL, Antunes AA, Salata LA. Immunohistochemical, tomographic, and histological study on onaly bone graft remodeling. Part III: allografts. Clin Oral Implants Res. 2013 Oct;24(10):1164-72. doi:10.1111/j.1600-0501.2012.02528.x. is possible the semi-quantitative scale of the IHC images can translated into quantitative number.

For immunohistochemical analyses on the bone graft area we used RANK, RANKL and OPG, which are widely used as bone tissue markers. Because that, we did only primary antibodies omission in the negative control according protocols from various studies published in the literature and can be proved by references shown below:

1.      Zuza EP, Garcia VG, Theodoro LH, Ervolino E, Favero LFV, Longo M, Ribeiro FS, Martins AT, Spolidorio LC, Zuanon JAS, de Toledo BEC, Pires JR. Influence of obesity on experimental periodontitis in rats: histopathological, histometric and immunohistochemical study.Clin Oral Investig. 2017 Sep 19

2.      Theodoro LH, Longo M, Novaes VCN, Miessi DMJ, Ferro-Alves ML, Ervolino E, de Almeida JM, Garcia VG. Low-level laser and antimicrobial photodynamic therapy on experimental periodontitis in rats submitted to chemotherapy by 5-fluorouracil. Support Care Cancer. 2017 May 9.

3.      Matheus HRErvolino EFaleiros PLNovaes VCNTheodoro LHGarcia VGde Almeida JM. Cisplatin chemotherapy impairs the peri-implant bone repair around titanium implants: An in vivo study in rats. J Clin Periodontol. 2017 Oct 1.

4.      Theodoro LH, Longo M, Ervolino E, Duque C, Ferro-Alves ML, Assem NZ, Louzada LM, Garcia VG. Effect of low-level laser therapy as an adjuvant in the treatment of periodontitis induced in rats subjected to 5-fluorouracil chemotherapy. J Periodont Res 2016; 51: 669–680.

5.      Garcia VG, Knoll LR, Longo M, Novaes VCN, Assem NZ, Ervolino E, de Toledo BEC, Theodoro LH. Effect of the probiotic Saccharomyces cerevisiae on ligature-induced periodontitis in rats. J Periodont Res 2015; 51: 26-37.

6.      Garcia VG, Longo M, Gualberto Júnior EC, Bosco AF, Nagata MJ, Ervolino E, Theodoro LH. Effect of the concentration of phenothiazine photosensitizers in antimicrobial photodynamic therapy on bone loss and the immune inflammatory response of induced periodontitis in rats. J Periodont Res 2014; 49:584–594.

Reviewer 2 Report

The manuscript topic is actual and the paper has merit. It could be attractive, adequate and interesting for the journal readers. However there are some point that authors should address in order to have a final more complete paper. Authors should underline the limitation of the value of the study, and the clinical and surgical implication of the presented study should be added. At this stage the paper seems to be directed to not clinical or surgeons readers. Please emphasize the clinical application of the study.
The limitation of an "animal study" should be underlined and need to be synthesized in a paragraph. 
....animal studies will only become more valid predictors of human reactions to exposures and treatments if there is substantial improvement in both their scientific methods as well as in more systematic review of the animal literature as it evolves. Systematic reviews of animal research, if they are used to inform the design of clinical trials, particularly with respect to appropriate drug dose, timing and other crucial aspects of the drug regimen, will further improve the predictability of animal research in human clinical trials....

Introduction section should highlights the clinical rationale of this paper. Otherwise the study seems to be directed to just scientist or researcher and not to clinicians. 

References are inadequate. Introduction section is poor. Some more references about the recent (2013-2019) CLINICAL reconstructive option just published have to be added. Please add the following ones:

Poli, Pier P et al. “Alveolar ridge augmentation with titanium mesh. A retrospective clinical study”  open dentistry journal vol. 8 148-58. 29 Sep. 2014, doi:10.2174/1874210601408010148

At the same time discussion is poor.
In the discussion section authors should compare the results of the present study with others one presented and published in the literature. Other important bone substitutes material and clinical studies are the following, please add:

Recombinant human bone morphogenetic protein type 2 application for a possible treatment of bisphosphonates-related osteonecrosis of the jaw.
Cicciù M, et al.
J Craniofac Surg. 2012 May;23(3):784-8. doi: 10.1097/SCS.0b013e31824dbdd4.

Author Response

Comments and Suggestions for Authors

The manuscript topic is actual and the paper has merit. It could be attractive, adequate and interesting for the journal readers. However there are some point that authors should address in order to have a final more complete paper. Authors should underline the limitation of the value of the study, and the clinical and surgical implication of the presented study should be added. At this stage the paper seems to be directed to not clinical or surgeons readers. Please emphasize the clinical application of the study.

Answer: The authors thank the reviewer for the considerations and hope that the answers have been satisfactory in clarifying the points raised.

The limitation of an "animal study" should be underlined and need to be synthesized in a paragraph. 
....animal studies will only become more valid predictors of human reactions to exposures and treatments if there is substantial improvement in both their scientific methods as well as in more systematic review of the animal literature as it evolves. Systematic reviews of animal research, if they are used to inform the design of clinical trials, particularly with respect to appropriate drug dose, timing and other crucial aspects of the drug regimen, will further improve the predictability of animal research in human clinical trials....
 Answer: This suggestion is already highlighted in the discussion.

Introduction section should highlights the clinical rationale of this paper. Otherwise the study seems to be directed to just scientist or researcher and not to clinicians. 
Answer: The authors agree that the present study is directed to scientists and researchers, and that the findings in this study aim to proposes hypotheses to better guide the understanding of biological events on the bone graft area in animals subjected to the conditions analyzed. This could inspire the scientific community so that further studies in this area are carried out with the aim of understanding the mechanisms of action and development of new techniques that could be applied in humans, so there is no direct clinical rationale of this paper.

References are inadequate. Introduction section is poor. Some more references about the recent (2013-2019) CLINICAL reconstructive option just published have to be added. Please add the following ones:

Answer: The most current references as recommended by the reviewer were added, however, the older references are classic and essential to support the study. Moreover, the present study aims to evaluate the influence of cigarette smoke inhalation on autogenous onlay bone graft area, either covered with a collagen membrane or not, in healthy and estrogen-deficient rats. So this study was not dedicated to the clinical outcomes of different treatment modalities of bone regeneration.

The authors added to the discussion: Elgali I, Omar O, Dahlin C, Thomsen P. Guided bone regeneration: materials and biological mechanisms revisited. Eur J Oral Sci. 2017 Oct;125(5):315-337. doi:10.1111/eos.12364, instead the one suggested by the reviewer for the introduction:

- Poli, Pier P et al. “Alveolar ridge augmentation with titanium mesh. A retrospective clinical study”  open dentistry journal vol. 8 148-58. 29 Sep. 2014, doi:10.2174/1874210601408010148.

At the same time discussion is poor.
In the discussion section authors should compare the results of the present study with others one presented and published in the literature. Other important bone substitutes material and clinical studies are the following, please add:

Recombinant human bone morphogenetic protein type 2 application for a possible treatment of bisphosphonates-related osteonecrosis of the jaw.
Cicciù M, et al. J Craniofac Surg. 2012 May;23(3):784-8. doi: 10.1097/SCS.0b013e31824dbdd4.

Answer: The authors acknowledge the suggestion of the article to include in the discussion but not added because it is not in congruence with the present study. The suggested article addresses the use of bisphosphonates and adjunctive therapy with recombinant human bone morphogenetic protein type 2 (rhBMP-2) associated with bone graft in humans, making it difficult to discuss and correlate with findings in our study.

Reviewer 3 Report

I have reviewed the manuscript “Influence of cigarette smoke inhalation on autogenous onlay bone graft area in rats with estrogen deficiency. A histomorphometric and immunohistochemistry study” submitted to IJMS. I found this manuscript interesting and fit well with in the scope of this journal. The authors have investigated the influence of smoking on bone, either covered with a collagen membrane or not, in healthy as well as estrogen-deficient rats. However, the manuscript needs some major improvements; there are a few suggestions that authors may consider to improve it further:

The title contains too many abbreviations; that may be unnecessary; please consider modifying to reduce abbreviations in title.

The abstract is not structured, very brief and poorly presented; the first line (26-27) look incomplete seems author wanted to describe the aim of the study. Please rephrase

The abstract is deficit in presentation of results section; authors should highlight key findings of the study. 

Introduction: line 42: “Periodontics and Implantodontics” should beperiodontics and implantodontics”.

Please cite reference for the statement lines 47-49.

Line 52: Guided Bone Regeneration (ROG): the standard abbreviation is GBR; and should replace ROG.

RANKL stands for Receptor activator of nuclear factor kappa-Β ligand? Please define

Regarding exposed to cigarette smoke inhalation; was that passive exposure or active; how that was controlled? The methodology section is well detailed otherwise.

There is no discussion about the limitations of the study should be detailed (almost not mentioned).

Please add a defined section of conclusions precisely briefing the findings of the study.

Author Response

Comments and Suggestions for Authors  

I have reviewed the manuscript “Influence of cigarette smoke inhalation on autogenous onlay bone graft area in rats with estrogen deficiency. A histomorphometric and immunohistochemistry study” submitted to IJMS. I found this manuscript interesting and fit well with in the scope of this journal. The authors have investigated the influence of smoking on bone, either covered with a collagen membrane or not, in healthy as well as estrogen-deficient rats. However, the manuscript needs some major improvements; there are a few suggestions that authors may consider to improve it further:

Answer: The authors thank the reviewer for the considerations and hope that the answers have been satisfactory in clarifying the points raised.

The title contains too many abbreviations; that may be unnecessary; please consider modifying to reduce abbreviations in title.

Answer: There are no abbreviations in the title.

The abstract is not structured, very brief and poorly presented; the first line (26-27) look incomplete seems author wanted to describe the aim of the study. Please rephrase. The abstract is deficit in presentation of results section; authors should highlight key findings of the study. 

Answer: According to the author's guide, the abstract should have a maximum of 200 words, which limits a better explanation of the interaction between groups and periods in the study. Below is a better summary formatting, however exceeding the number of words for this journal topic.

“Purpose: The present study aims to evaluate the influence of cigarette smoke inhalation on autogenous onlay bone graft area, either covered with a collagen membrane or not, in healthy and estrogen-deficient rats through histomorphometry and immunohistochemistry. Materials and Methods: Sixty female rats (Wistar), weighing 250-300 g, have been randomly divided and allocated into a groups (either exposed to cigarette smoke inhalation or not, ovariectomized and SHAM). Fifteen days afterwards, the test group underwent cigarette smoke inhalation. Sixty days after exposition, autogenous bone grafting was only performed on all right hemimandibles, and the left ones underwent autogenous onlay bone grafting with the collagen membrane (BioGide®). The graft was harvested from the parietal bone and attached to the animals’ jaws (right and left). They were euthanized at 21, 45 and 60 days after grafting. Histological measurements and immunohistochemical analyzes were performed, whose results were submitted to a statistical analysis. Results: The addition of a collagen membrane to the bone graft proved more efficient in preserving graft area if compared to the graft area without a collagen membrane and the one associated with cigarette smoke inhalation at 21 and 60 days, respectively (COM x TO, p=0.0381 and TO x COM, p=0.0192). Cigarette smoke inhalation combined with ovariectomy promoted a significant reduction of autogenous graft area at 21 and 60 days. At 45 days, no statistically significant results were observed. In the immunohistochemical analysis, the test subgroups received moderate and intense immunolabeling at 21 days for RANKL (TOM, p=0.0017 and TO, p=0.0381).  For OPG, intense immunolabeling was observed in most subgroups under analysis at 60 days. Conclusion: The smoking inhalation promoted resorption on the autogenous onlay bone graft, mainly when associated with ovariectomy. Furthermore, when associated with the collagen membrane, a lower resorption rate is observed if compared to the absence of the membrane.”

Introduction: line 42: “Periodontics and Implantodontics” should be “periodontics and implantodontics”.

Answer: The text was changed.

Please cite reference for the statement lines 47-49.

Answer: The references were added to the text.

Line 52: Guided Bone Regeneration (ROG): the standard abbreviation is GBR; and should replace ROG.

Answer: The abbreviation was replaced.

RANKL stands for Receptor activator of nuclear factor kappa-Β ligand? Please define

Answer: The text was changed.

Regarding exposed to cigarette smoke inhalation; was that passive exposure or active; how that was controlled? The methodology section is well detailed otherwise.

Answer: The exposure was passive and controlled through the use of 10 cigarettes at a time (concentration of 1.3 mg of nicotine, 16.5 mg of tar and 15.2 mg of carbon monoxide - MINISTER King size Unique - Souza Cruz, Rio de Janeiro, Brazil) 3 times a day for 8 minutes each, as cited in the text.

There is no discussion about the limitations of the study should be detailed (almost not mentioned).

Answer: The text was changed and a references were added.

Pearce AI, Richards RG, Milz S, Schneider E, Pearce SG. Animal models for implant biomaterial research in bone: a review. Eur Cell Mater. 2007 Mar 2;13:1-10. Review.

Elgali I, Omar O, Dahlin C, Thomsen P. Guided bone regeneration: materials and biological mechanisms revisited. Eur J Oral Sci. 2017 Oct;125(5):315-337. doi: 10.1111/eos.12364. Epub 2017 Aug 19. Review.

Please add a defined section of conclusions precisely briefing the findings of the study.

Answer: The text was changed.
